# Clinical Spectrum of *LMNA*-Associated Type 2 Familial Partial Lipodystrophy: A Systematic Review

**DOI:** 10.3390/cells12050725

**Published:** 2023-02-24

**Authors:** Antia Fernandez-Pombo, Everardo Josue Diaz-Lopez, Ana I. Castro, Sofia Sanchez-Iglesias, Silvia Cobelo-Gomez, Teresa Prado-Moraña, David Araujo-Vilar

**Affiliations:** 1UETeM-Molecular Pathology Group, Department of Psychiatry, Radiology, Public Health, Nursing and Medicine, IDIS-CIMUS, University of Santiago de Compostela, 15706 Santiago de Compostela, Spain; 2Division of Endocrinology and Nutrition, University Clinical Hospital of Santiago de Compostela, 15706 Santiago de Compostela, Spain; 3CIBER Fisiopatología de la Obesidad y la Nutrición (CIBERobn), 28029 Madrid, Spain

**Keywords:** *LMNA*, Dunnigan disease, type 2 familial partial lipodystrophy, laminopathies, FPLD2

## Abstract

Type 2 familial partial lipodystrophy (FPLD2) is a laminopathic lipodystrophy due to pathogenic variants in the *LMNA* gene. Its rarity implies that it is not well-known. The aim of this review was to explore the published data regarding the clinical characterisation of this syndrome in order to better describe FPLD2. For this purpose, a systematic review through a search on PubMed until December 2022 was conducted and the references of the retrieved articles were also screened. A total of 113 articles were included. FPLD2 is characterised by the loss of fat starting around puberty in women, affecting limbs and trunk, and its accumulation in the face, neck and abdominal viscera. This adipose tissue dysfunction conditions the development of metabolic complications associated with insulin resistance, such as diabetes, dyslipidaemia, fatty liver disease, cardiovascular disease, and reproductive disorders. However, a great degree of phenotypical variability has been described. Therapeutic approaches are directed towards the associated comorbidities, and recent treatment modalities have been explored. A comprehensive comparison between FPLD2 and other FPLD subtypes can also be found in the present review. This review aimed to contribute towards augmenting knowledge of the natural history of FPLD2 by bringing together the main clinical research in this field.

## 1. Introduction

The *LMNA* gene codifies for lamin A/C, which is an intermediate filament protein present in the nuclear lamina and an important determinant of nuclear and cellular architecture. The correct processing of the A-type lamins is essential for the prevention of several disorders [1]. In this regard, prelamin A is farnesylated, methylated and processed by zinc metalloprotease to form the mature lamin A (Figure 1).

Thus, there are a number of diseases caused by pathogenic variants in the *LMNA* gene (familial partial lipodystrophy type 2 [FPLD2] or Dunnigan disease, Hutchinson-Gilford progeria syndrome, mandibuloacral dysplasia type A, Emery Dreifuss muscular dystrophy, Limb-Girdle muscular dystrophy or atypical progeroid syndrome, among others) or other genes that influence this lamin processing (such as *ZMPSTE24* gene variants in mandibuloacral dysplasia type B or in restrictive dermopathy type 1) or in genes that influence its proper functioning on chromatin (such as *BANF1* gene variants in Néstor-Guillermo progeria syndrome). These disorders are known as laminopathies, which mainly affect mesenchymal tissues (muscle, adipose tissue and bone), although some affect the nervous system (Figure 1) [1,2,3].

FPLD2 also belongs to a bigger family of FPLD syndromes that include a set of Mendelian disorders due to variants in different genes related to adipogenesis and lipogenesis and which share subcutaneous fat loss from the limbs and gluteal region, in addition to the variable regional accumulation of excess fat [4,5,6]. Due to adipose tissue dysfunction, ectopic fat accumulation and, consequently, insulin resistance, they also share several metabolic abnormalities and comorbidities. However, there is a great degree of phenotypical variability between the different FPLD disorders. To date, seven subtypes and another four unclassified variants of FPLD have been described, with FPLD2 and FPLD1 being the most frequent [7].

The extreme rarity of lipodystrophy syndromes such as FPLD2 implies that they are not well-known. In fact, the knowledge we currently have about the clinical characterisation of this disease, the associated comorbidities and its natural course is mainly based on studies limited to small samples. In addition, among the most fascinating traits of this disorder, as in the rest of laminopathies, are its complex genotype-phenotype associations and its clinical heterogeneity [2]. 

Thus, this review aimed to collect and summarise the published data regarding the classical and atypical clinical features of Dunnigan disease and its associated comorbidities, as well as the differential diagnosis with other FPLD subtypes, in order to contribute to the understanding of this disorder.

## 2. Materials and Methods

### 2.1. Search Strategy

To show the current knowledge about the clinical characteristics, organ abnormalities and associated comorbidities of FPLD2, a systematic review based on the Preferred Reporting Items for Systematic Reviews and Meta-Analyses (PRISMA) protocol was conducted [8]. The protocol for this systematic review is also published in the Open Science Forum registries and can be accessed using the following: https://osf.io/d453h (accessed on 15 February 2023). For this purpose, a search on PubMed was performed using the terms “Dunnigan” OR “familial partial lipodystrophy” OR “FPLD”. The database was searched up to December 2022, without a time limitation and without language restrictions. The references of the retrieved articles were also screened to identify additional studies and expand our search (Figure 2).

### 2.2. Study Selection

A total of 788 articles were identified through this database search. Studies that report clinical and/or biochemical data on patients with FPLD2 were eligible for inclusion in the current review. The main exclusion criteria were (a) articles not fulfilling the topic of interest of this review; (b) review articles, meta-analyses, individual case reports, editorials or comments; and (c) studies not conducted on humans. Two authors independently reviewed the abstracts of the retrieved articles, applying the previously-mentioned inclusion and exclusion criteria and, subsequently reviewed the full text of the studies to determine their final inclusion. 

### 2.3. Identification and Inclusion of Studies

From the 788 articles initially found, 454 were excluded taking into account that they did not fulfil the topic of interest of this review, two were excluded because they were published errata and 66 were excluded because they were focused on other FPLD subtypes or laminopathies, other than Dunnigan disease. During the screening and eligibility processes, 79 review articles or meta-analyses, 71 individual case reports, eight letters to the editor and one comment were also excluded. In addition, a total of 6 studies were finally included after the screening of the references of the initial pool of retrieved articles. Thus, finally, 113 articles were included in the current review. The flow diagram regarding the research strategy can be seen in Figure 2. 

## 3. Results

### 3.1. Prevalence

Since the first report of Dunnigan disease in 1974 [9], several cases have been described over the years. Although it is difficult to speak of prevalence in a rare disorder, in 2017, taking into account database and literature searches, a prevalence of 1.7–2.8 cases per million was determined for partial lipodystrophy (both FPLD and acquired partial lipodystrophy) [10]. 

With the aim of evaluating the prevalence of missense variants in the *LMNA* gene exclusively, a DNA sequencing data extraction was performed from 60,706 unrelated individuals in the Exome Aggregation Consortium (ExAC) database. Allele frequencies ranged from 1-per-100,000 to 1-per-1,000 and were primarily heterozygous (only two were homozygous [p.S625C and p.R644C]) [11].

In a more recent analysis, looking for clinical and molecular lipodystrophy diagnostic codes, also using the ExAC database, a pathogenic variant carrier prevalence for autosomal dominant FPLD of ~1 in 7588 individuals was estimated. What is more striking is that additional pathogenic variants in the *LMNA* gene (such as the p.R482Q variant associated with FPLD2) were also found in patients with metabolic abnormalities in the lipodystrophy spectrum, but without a clinical diagnosis [12]. This only confirms that lipodystrophy syndromes such as Dunnigan disease are underdiagnosed and underestimated conditions, poorly understood by physicians outside the specialist treatment centre setting.

Thus, there is definitely great difficulty in making a realistic estimate of a specific subgroup of rare and heterogeneous diseases, and only large and prolonged registries will make it possible to present a more objective picture of the real data.

### 3.2. Clinical Characteristics

As for most FPLD subtypes (except for types 5, 6 and those associated with *PCYT1A* and *MFN2* genes), the reported transmission of FPLD2 supports an autosomal dominant mode of inheritance [13,14,15]. However, this disorder can actually be considered a co-dominant disease, taking into account that homozygous cases have also been described [16].

It is classically characterised by the loss of fat from the trunk, buttocks and upper and lower limbs, in addition to fat accumulation in the face, neck and supraclavicular fossae, giving a Cushingoid appearance [13,15,17,18] (Figure 3). In this sense, in vivo morpho-functional assessment of fat depots in the neck area of FPLD2 patients by PET/CT analysis exhibited the absence of brown adipose tissue activity, therefore showing that failure in adipose tissue browning may be a contributor to disease [19]. Trunk lipoatrophy also contributes to the appearance of labial hypertrophy [15]. Lipomas are likewise a common characteristic of these patients, with a reported prevalence of 20%, even in lipoatrophic areas [20]. Along with this abnormal fat distribution, muscular hypertrophy and myalgias are also recurrent features in these patients [13,21,22,23].

Other characteristic features of these patients are the presence of very prominent peripheral veins due to the lack of subcutaneous fat [24], the presence of signs of insulin resistance such as acanthosis nigricans and acrochordons [16,25,26], as well as signs of hyperandrogenism in affected women, such as hirsutism [13,23]. 

Although the onset of the phenotype has been described in some cases in childhood, with other signs and comorbidities occurring later in life [27], in most cases, in this specific FPLD subtype, lipodystrophy usually appears progressively around puberty in women, and later in men [13,14,23,28].

Patients sometimes admit to having several difficulties coming to terms with their appearance, relating to physical discomfort and psychological distress, especially in the case of women, which may lead them to consider the possibility of undergoing cosmetic surgery [14].

Clinical research has also revealed that the expression and severity of the phenotype may be markedly dependent on sex, with women being more affected. On the contrary, in men, the clinical diagnosis of FPLD can be more problematic, taking into account the less-apparent lipodystrophy phenotype in most cases and its later onset (Figure 3), leading to delays in the diagnosis [13,26]. In fact, men are usually diagnosed after their female relatives.

When comparing heterozygous and homozygous patients, more severe lipoatrophy was also found in the latter. Thus, in a recent study conducted on 65 patients carrying the monoallelic *LMNA* p.(Thr655Asnfs*49) variant and 13 carrying the same biallelic variant, while 54% of heterozygous subjects were diagnosed during family screening, 69% of homozygous patients were diagnosed due to medical complications, all of them presenting obvious clinical features of lipodystrophy [16]. In this sense, compound heterozygosity in the *LMNA* gene was also found to be associated with a relatively more severe FPLD2 phenotype [29].

It has also been reported that most FPLD2 patients with the previously described classic phenotype are those who harbour heterozygous missense variants affecting arginine at codon 482 in exon 8 of the *LMNA* gene, while those presenting with other *LMNA* variants are considered to have atypical FPLD2. Thus, patients with the p.(R644C), the p.(R582H) or the p.(R582C) variants in exon 11, patients with the p.(T528M) variant in exon 9, patients with the p.(D47N) variant in exon 1, or even patients with the p.(R471G) variant in exon 8 of the *LMNA* gene, were shown, in previous studies, to have a milder phenotype, with less severe loss of fat, occurring at an older age in some cases. In contrast, other cases with the p.(R349W) variant in exon 6 showed a more remarkable fat loss, even in the face [5,30,31,32,33,34,35,36]. Other distinctive features found in subjects with non-codon 482 FPLD2 were the absence of prominent labia majora, the lack of acanthosis nigricans and hirsutism [5] and the presence of frequent muscle signs such as myalgias or weakness [37]. In addition, other cases of FPLD2 with other heterozygous pathogenic variants, such as p.(R541P) or p.(K486E), were reported to have a more complex phenotype that may more closely resemble generalised lipodystrophy [38]. This clinical heterogeneity suggests the possible influence of other factors, such as environment, which might induce epigenetic changes or the presence of single nucleotide polymorphisms in *LMNA* that could modulate the clinical expressivity of this disorder.

### 3.3. Body Composition

Throughout the literature, the body composition of patients with FPLD2 has mainly been evaluated through dual-energy X-ray absorptiometry (DXA) (Figure 4) or magnetic resonance imaging (MRI), although some isolated studies have used bioelectrical impedance analysis (BIA) for this purpose. These body composition techniques have been shown to help diagnosis by revealing even subclinical changes in fat deposition suggestive of lipodystrophy [37,39]. Through these techniques, the previously mentioned clinical findings of the near-total absence of adipose tissue in the limbs and truncal area (without reduction in intraabdominal and intrathoracic fat), along with an excess of adipose tissue in the face, neck, chin, axillae, and labia majora have been confirmed [4,30,40,41,42]. In addition, using the Dixon method of MRI, not only was quantified fat liver observed to be high in FPLD2 subjects, but pancreatic fat was also greater in this population. Furthermore, a positive relationship was demonstrated between these specific fat contents and metabolic parameters such as glycated haemoglobin (HbA1c) or triglycerides [41,43].

Although computed tomography and MRI remain the reference standards for assessing visceral adipose tissue (VAT) and subcutaneous abdominal adipose tissue (SAT) distribution, these are expensive and limited techniques with certain operative contraindications. Thus, the determination of possible correlations between MRI data of truncal adiposity and simpler clinical measures, such as BIA-derived data in FPLD women, could be of use in the assessment of body composition in these subjects. In this sense, a significant correlation between mid-thigh fat percentage by MRI and BIA has been observed [44], and the measurement of total fat percentage most strongly correlated with both VAT and SAT [45].

Furthermore, in line with previously described clinical findings, women with atypical FPLD showed less severe loss of adipose tissue, evaluated via MRI, from the limbs and trunk, and particularly from the gluteal region and proximal thighs than women with typical FPLD [5,33]. Despite this loss of fat, periarticular adipose tissue was found to be preserved in the lower limbs of patients with Dunnigan disease [46]. In addition, several studies compared the body composition of FPLD2 subjects with other FPLD subtypes, such as that associated with pathogenic variants in the *PPARG* gene. In these studies, patients with FPLD3 showed higher limb fat, with preserved truncal fat mass and increased skinfold thickness in the thigh, calf, triceps, and biceps, along with higher levels of leptin than FPLD2 subjects [36,47,48,49]. Although FPLD3 patients are, therefore, considered to have milder lipodystrophy, it has been observed that they develop more severe metabolic complications [47], suggesting that the severity of these metabolic disturbances may not only be associated with the extent of fat loss.

On the other hand, several adipose tissue cut-offs and indexes have been proposed in the literature to help guide the diagnosis of Dunnigan disease. Thus, lower-limb fat measured by DXA in female children with FPLD2 was initially found to be below or equal to the 1st percentile for NHANES [28]. In a subsequent study, it was also determined that lower-limb fat percentage below the 1st percentile according to DXA may direct the diagnosis of FPLD2 in women (with a specificity of 0.995 and a sensitivity of 1.0), especially if there are concomitant metabolic complications, and, therefore, genetic testing should be carried out in these cases [50]. Fat Mass Ratio (FMR), defined as the ratio between the trunk and lower-limb fat mass through DXA, was likewise proposed. It showed a greater value for FPLD2 subjects in comparison with controls as well as improved accuracy for evaluating these patients with a cut-off point of 1.2 [51]. Another index proposed as an objective measurement capable of determining fat excess in these patients was the body adiposity index (BAI), calculated as (hip circumference/height1.5)-18. Thus, BAI was found to be lower in FPLD2 in comparison with age and BMI-matched healthy individuals and presented a more significant correlation with parameters such as total fat percentage and fat mass, as well as with leptin levels, than BMI [52].

DXA may provide not only quantitative but also qualitative information. In this sense, a method was recently described which may be useful in the diagnosis of lipodystrophy syndromes such as FPLD2 through the reconstruction of DXA images using a colour-coded representation that highlights only adipose tissue (“fat shadows”). This method was able to differentiate FPLD from control subjects with 85% sensitivity and 96% specificity. In addition, the identification of the “Dunnigan sign” (hypertrophy of mons pubis fat surrounded by subcutaneous lipoatrophy) is also helpful in recognising subjects with FLPD2, which can easily be acknowledged by this specific method [53].

As far as skeletal muscle is concerned, in comparison with healthy women matched for age and BMI, patients with FPLD2 showed greater volume in the thigh, calf and psoas muscles, as well as increased arm and leg muscle masses when measured via DXA. In addition, insulin sensitivity was shown to be negatively correlated to calf muscle volume [54]. However, in a recent study, despite this increased muscularity, FPLD2 subjects did not demonstrate increased muscle strength and even showed earlier fatigue on chest-press exercise. Furthermore, the increase in skeletal muscle was found to be likely due to reduced muscle protein degradation rather than to greater protein synthesis, with impaired mitochondrial function playing a relevant role in this dysfunction [55]. 

On the other hand, bone can also be assessed through body composition techniques such as DXA. Laminopathies such as Hutchinson-Gilford progeria syndrome, mandibuloacral dysplasia, *LMNA*-associated atypical progeroid syndrome, Néstor-Guillermo progeria syndrome, restrictive dermopathy or lethal foetal akinesia are characterised by bone alterations, suggesting that lamin A/C could play an important role in the pathogeny of the loss of bone mass related to ageing [56,57,58,59]. However, in the case of FPLD2, no differences in bone mineral density evaluated via DXA were found compared to non-lipodystrophic obese women or women with FPLD1 [60]. Nevertheless, in another study, several patients with FPLD showed non-specific degenerative radiographic abnormalities such as osteoarthritis or calcific tendonitis and/or osteochondrosis [61].

### 3.4. Comorbidities and Organ Abnormalities

Most lipodystrophy syndromes such as FPLD2 are characterised by the presence of insulin resistance and, consequently, by a variable degree of metabolic dysfunction, with diabetes, dyslipidaemia, fatty liver disease, cardiovascular disease, and reproductive dysfunction. In fact, it is known that patients with different FPLD subtypes with similar truncal mass to subjects with non-FPLD obesity have worse metabolic profiles [62]. In addition to several clinical characteristics, Table 1 shows the main reported comorbidities of FPLD2 as well as specific differential features and comorbidities of other FPLD syndromes for their differential diagnosis, taking into account the great similarities of these lipodystrophy syndromes [63,64,65,66,67,68,69,70,71,72,73,74,75,76,77,78,79,80,81,82,83,84,85,86,87].

It has been reported that comorbidities in Dunnigan disease develop after age 10, which is why it has been proposed that while clinical review and dietetic support are beneficial for children with this disorder, formal screening for organ abnormalities before the age of 10 may not be of benefit. However, this recommendation has to be taken with caution considering the anticipation phenomenon recently described for this population, with the occurrence of metabolic complications such as diabetes and hypertriglyceridaemia at an earlier age across generations [28,88,89].

#### 3.4.1. Adipokine Disturbance

Reduced synthesis and secretion of adipocyte-specific proteins may be related to the metabolic complications of lipodystrophy. Thus, several studies have evaluated the differences between serum leptin or adiponectin levels, among other adipokines, between FPLD2 subjects and matched controls or patients with generalised lipodystrophy. In this sense, FPLD2 patients, as expected, showed significantly lower plasma levels of adiponectin and leptin in comparison with healthy controls [30,90,91,92]. In addition, adiponectin levels were found to be inversely correlated with insulin resistance and fasting serum triglycerides and positively correlated with serum HDL cholesterol levels [90,93,94]. Nevertheless, while generalised lipodystrophy is usually characterised by very low or undetectable levels of leptin, patients with partial lipodystrophy, such as FPLD2, may exhibit leptin concentrations ranging from low to normal values [93]. In addition, there are no standardised cut-offs defined for leptin levels to confirm or exclude the diagnosis of lipodystrophy. 

More contradictory results have been found in relation to other adipokines such as TNF-α, IL-1β, IL-6 or IL-10. While in some studies, they were shown to be increased [30,95], IL-6 was also found to be decreased in others [90]. Retinol-binding protein 4 (RBP4) is also considered one of the many metabolic biomarkers identified in recent decades, with FPLD2 patients showing significantly lower levels of this protein compared with healthy controls, as well as a negative correlation with blood pressure [96].

#### 3.4.2. Insulin Resistance and Diabetes

Throughout the literature, patients with Dunnigan disease showed increased fasting glucose, increased HbA1c values, high plasma insulin and higher insulin resistance index (HOMA-IR) than healthy controls [16,24,95,97,98]. After glucose intake, FPLD2 patients showed an excessive insulin response and revealed insulin resistance [94,97,99,100], as well as a decreased response of plasma glucose to exogenously administered insulin [99].

The prevalence of diabetes in Dunnigan disease ranged from 28% to 51% [16,45,94,101,102,103,104]. However, when separated according to gender, the prevalence of diabetes increased to 54% in women and decreased to 17% in men [5,101]. In addition, compared to heterozygous patients, homozygous subjects tended to develop diabetes more frequently and earlier [16]. The prevalence of diabetes is higher in the latter compared to patients with FPLD3 [47].

#### 3.4.3. Lipid Metabolism and Pancreatitis

The prevalence of dyslipidaemia was described to be clearly much higher in patients with FPLD2 than in healthy-matched controls, ranging from 59% to 89% throughout the different studies that assessed lipid metabolism in this population [16,26,47,105]. The lipid profile consisted fundamentally of high triglyceride levels and low HDL cholesterol, which was more pronounced in women and in homozygous subjects [5,16,28,92,95,106,107]. In fact, dyslipidaemia is considered one of the earliest metabolic disturbances that occur in patients with *LMNA* variants [27]. However, comparing its prevalence with that of patients with FPLD3, it was higher in the latter [47].

Severe and refractory hypertriglyceridaemia has been related to acute pancreatitis [108]. The prevalence of the latter has been previously assessed in a Canadian cohort of 74 FPLD2 subjects, ranging from 4.1% in the case of hospitalisation for acute pancreatitis to 10.7% in the case of patients with both diabetes and pancreatitis, concluding that the risk of hypertriglyceridaemia-associated acute pancreatitis in FPLD2 may be substantial if diabetes is present [104]. In another study, when analysing gender differences in eight families with FPLD2, serum triglycerides were markedly elevated in affected women, and the prevalence of acute pancreatitis events was 29% compared to 0% in men [5]. It is important to highlight that the consumption of oral oestrogens is contraindicated in these patients due to the risk of exacerbation of hypertriglyceridaemia and the concomitant risk of developing acute pancreatitis. In addition, throughout the literature, some patients have also shown eruptive xanthomas and recurrent episodes of acute pancreatitis [104,106], and autopsy findings revealed severe amyloidosis of the pancreatic islets in one woman with the R482Q variant [108].

Variable expressivity related to different lipid profiles in patients with FPLD2 occurs when comparing typical with atypical FPLD. While in some studies, subjects with atypical FPLD2 showed lower serum triglyceride and HDL cholesterol concentrations [5], in another analysis, hypertriglyceridaemia was similar in patients with codon 482- and non-codon 482-associated variants [37]. This variable expressivity even occurs within exon 8 of the *LMNA* gene [109]. Thus, in a population of 32 subjects with the R482 and 15 subjects with the N466 variants, the latter group of patients showed higher serum triglyceride levels and, accordingly, acute pancreatitis was only present in these subjects (20%). On the contrary, patients with the R482 variant showed higher LDL cholesterol concentrations, while no differences were observed regarding non-HDL cholesterol between both groups [110].

#### 3.4.4. Cardiovascular Disease

Another associated metabolic complication related to insulin resistance in these patients is cardiovascular disease. To date, several studies have described an increased prevalence of cardiac manifestations in FPLD2 patients, such as cardiac hypertrophy [23,111], atrioventricular conduction defects and heart failure due to ventricular dilatation [112,113]. Early atherosclerosis occurring before the age of 45 has been reported [114,115,116], and a higher prevalence in women in comparison with men has also been described [5]. In addition, rhythm disturbances, such as atrial fibrillation or flutter, are likewise well-known cardiac abnormalities described for this population [112,113,117]. In fact, a recent study conducted on 122 FPLD patients, of whom a total of 66 presented pathogenic variants in the *LMNA* gene, showed that arrhythmias were significantly more frequent in patients with Dunnigan disease, with an odds ratio of 3.77 (95% CI: 1.45–9.83) and with a shorter time to the first rhythm disturbance in comparison with patients with other FPLD subtypes [117]. In another study, epicardial adipose tissue assessed with transthoracic echocardiography was also found to be higher in subjects with FPLD2 than in age and BMI-matched controls. However, this was not correlated with anthropometric or DXA measurements or with metabolic biochemical parameters [118]. In addition, along with other metabolic disturbances, hypertension has also been described for this population, with a prevalence of around 30–41% of FPLD2 subjects, with no differences regarding gender [5,26,47,103].

On the other hand, when comparing patients with heterozygous *LMNA* variants at the 482nd codon and patients with non-codon 482 *LMNA* variants, the latter group of subjects was shown to be more likely to have cardiac events such as arrhythmias [117,119]. If heterozygous and homozygous patients are compared, homozygous subjects harbouring the p.(Thr655Asnfs*49) variant in exon 11 of the *LMNA* gene presented more overlapping syndromes with severe cardiac phenotypes (early atheroma with coronary heart disease, conduction or rhythm disorders, dilated cardiomyopathy, heart failure with left ventricle lowered ejection fraction and an earlier onset of these cardiac manifestations in comparison with heterozygous subjects with the same variant) [16,120].

Thus, these results provide evidence of the association of multisystem and overlapping syndromes in FPLD2 with an increased frequency of cardiac abnormalities, therefore highlighting the need for vigilant cardiac monitoring in patients with Dunnigan disease.

#### 3.4.5. Liver Abnormalities

Fatty liver disease is another frequent condition in subjects with FPLD2, mainly due to inadequate adipose storage and ectopic fat accumulation that impairs insulin signalling and other cellular functions. Ectopic lipid storage in insulin-targeted organs, including the liver, promotes insulin resistance and its complications, including non-alcoholic fatty liver disease (NAFLD), which was reported in the literature in up to 83% of FPLD2 cases [16]. In addition, in an analysis conducted on 14 women with FPLD2 (13 with the R482W variant and one with the R644C variant), all of them exhibited hepatic steatosis in MRI [40]. Hepatic steatosis, according to ultrasound, was also present in 15 out of 18 subjects with FPLD-causing variants in *LMNA*, with nine of these subjects showing elevated serum aminotransferase activities [105]. Furthermore, in another study conducted on 23 patients with lipodystrophy syndrome (seven of them with heterozygous pathogenic variants in *LMNA*) and NAFLD, who underwent liver biopsy, it was shown that 22 met histopathological criteria for non-alcoholic steatohepatitis (NASH) [43].

#### 3.4.6. Gynaecological and Obstetric Disorders

Only a few studies have assessed gynecological and obstetric abnormalities in patients with FPLD2. In this sense, a frequent condition found in these individuals is polycystic ovary syndrome (PCOS), with an estimated prevalence throughout the literature of approximately 16–54% [103,121,122,123], possibly due to the condition of severe insulin resistance, ovarian lipotoxicity due to free fatty acid accumulation and intra-abdominal fat accumulation. In fact, in a report of two cases with the R482Q variant in the *LMNA* gene, the amelioration of insulin resistance and hyperinsulinaemia with the use of pioglitazone resulted in the normalisation of ultrasound ovarian morphology. However, in a previous analysis, PCOS appeared to be more prevalent in *PPARG* variant carriers than in subjects with *LMNA* variants [124]. Other gynaecological/obstetric complications reported with greater frequency in FPLD2 subjects than in the general population are infertility (28%), gestational diabetes (36%), miscarriage (50%) and eclampsia and foetal death (14%) [121].

#### 3.4.7. Neuromuscular Disturbances

Apart from the previously mentioned muscle hypertrophy and myalgias [14,22], multiple nerve entrapment syndromes have also been reported in FPLD2. Whereas in a skeletal muscle histology study, non-specific myopathic changes were observed along with marked type one and two muscle fibre hypertrophy, in sural nerve biopsies, numerous paranodal myelin swellings were found. Thus, it was concluded that the myopathy and neuropathy associated with FPLD2 are clearly distinct from other laminopathies [21].

#### 3.4.8. Renal Disease

Renal involvement (proteinuria and progression to renal failure) has also been reported in the course of FPLD2 [125], with poorly controlled diabetes driving these alterations in only some cases. In addition, in a case with the R644C variant in the *LMNA* gene and in four members of a large pedigree with the pathogenic R349W variant in the same gene, focal segmental glomerulosclerosis was present, with a variable degree of renal dysfunction and proteinuria, suggesting, therefore, a possible role of *LMNA* in the maintenance of glomerular function and structure [126,127].

#### 3.4.9. Other Organ Abnormalities

Thyroid disease is not considered a common organ abnormality in patients with Dunnigan disease. However, it was shown to be more frequent in subjects with the p.(Thr655Asnfs*49) variant in the *LMNA* gene than in control subjects [16]. In addition, in a previous report, subcutaneous adipose tissue in the thigh of subjects with FPLD2 showed lower expression of thyroid hormone transporters and higher expression of iodothyronine deiodinases than abdominal subcutaneous adipose tissue, suggesting that changes in thyroid hormone metabolism may occur in areas with lipoatrophy in these patients [128].

Contrary to other laminopathies associated with premature ageing, taking into account the role of A-type lamins in the regulation of cell proliferation, to the best of our knowledge, Dunnigan disease has not been associated with an increased risk of malignancy. However, the development of hypopharyngeal squamous cell carcinoma in the absence of any other risk factors for head and neck cancer has been reported in two biological sisters with the R349W variant in the *LMNA* gene [129].

### 3.5. Mortality

Only a few studies have analysed the survival of FPLD2 subjects. In an international chart review study, which investigated overall survival in a cohort of 149 patients with partial lipodystrophy and 81 patients with generalised lipodystrophy from the USA, Turkey, and Brazil, the mean time to death was 66.6 ± 1.0 years for patients with partial lipodystrophy [130]. Among the eight patients with partial lipodystrophy who died during follow-up, four presented FPLD2. In another study that addressed mortality in a metreleptin-naïve cohort (103 subjects with either partial or generalised lipodystrophy), there was a total of nine deaths among patients with generalised lipodystrophy and three deaths among patients with partial lipodystrophy (all of them FPLD) [131]. Although cardiovascular events were reported to be one of the most frequent potential contributing factors to death in the overall cohorts, none of the studies made specific reference to the cause of mortality in patients with *LMNA*-associated variants.

### 3.6. Treatment

At present, there is no cure for this condition. Therapeutic approaches should be directed towards the associated comorbidities. 

Diet, along with physical exercise, is an integral part of the treatment plan of these patients. However, in a study conducted on 14 women with FPLD2, dietary intake was evaluated, with these patients showing a lower energy, lipid and carbohydrate intake and a larger protein intake in comparison with a healthy-matched control group of individuals. In addition, in this study, it was reported that 78% of FPLD2 women performed insufficient physical activity. They also showed higher HbA1c and triglycerides. However, no correlation between dietary intake and biochemical parameters was found [132]. According to the Multisociety Lipodystrophy Guidelines [133], the recommended diet for these patients consists of a 50–60% carbohydrate, 20–30% fat, and approximately 20% protein diet. Particularly in adults, a hypocaloric diet could be beneficial, with particular emphasis on the reduction of simple carbohydrates and fat and alcohol abstinence. Regarding physical exercise, this should be encouraged once cardiac disorders have been ruled out, albeit with caution, as heart disease could appear later in life. Thus, frequent cardiological check-ups would be advisable. 

Although, to date, metformin continues to be the therapy of choice for the initial treatment of diabetes [133], other oral or subcutaneous hypoglycaemic agents have been specifically assessed in patients with FPLD2. This is the case of thiazolidinediones, which selectively stimulate PPARG and have been shown to reduce fasting glucose levels, free fatty acid concentrations and liver transaminases, presumably through adipose tissue expansion in these subjects [134]. Considering the increased levels of dipeptidyl peptidase-4 found in another study, glucagon-like peptide-1 receptor agonists may also be taken into account when considering glucose-lowering therapy in this population [135]. However, the use of these specific drugs in subjects at increased risk of pancreatitis should be taken with caution. 

Dyslipidaemia in these patients should be managed in agreement with the current guidelines for the general population, with statins being the therapy of choice and fibrates and long-chain omega-3 fatty acids reserved for severe hypertriglyceridaemia (>500 mg/dL) [133]. In this sense, with the main objective of evaluating the reduction of triglyceride levels in FPLD subjects, new drugs have been investigated. In this sense, the efficacy and safety of volanesorsen, an antisense inhibitor of apolipoprotein C-III, has been evaluated in a 52-week phase 2/3 study, which randomised 1:1 to weekly administration of volanesorsen (300 mg) or placebo. This study showed an 88% reduction in triglycerides after 3 months, as well as a significant reduction in a hepatic fat fraction in 40 subjects with FPLD [136]. Vupanorsen, an inhibitor of ANGPTL3, was also studied in a recent analysis conducted on four patients with FPLD (two with pathogenic variants in the *LMNA* gene), severe hypertriglyceridaemia and hepatic steatosis. A reduction of triglyceride fasting levels of 59.9%, as well as in other lipoproteins, was observed [137].

In addition, in 2007, the first specific study assessing the effects of long-term recombinant leptin replacement in six patients with Dunnigan disease was carried out. In this analysis, triglyceride levels were reduced by 65% at 4 months and significantly reduced at 12 months for five patients. A reduction in total cholesterol as well as an improvement in insulin sensitivity and fasting levels after 12 months of treatment, were also observed without significant changes in HbA1c [138]. Other studies conducted on subjects with FPLD were subsequently carried out, with significant reductions in triglyceride levels also observed. As far as glucose metabolism is concerned, while in some analyses, the absence of significant changes in HbA1c has been confirmed for this population [139,140], in others, improvements in glycaemic control, insulin sensitivity and insulin secretion were reported [141,142]. On the other hand, in two subjects with FPLD2 belonging to a larger cohort of 10 patients with either generalised or partial lipodystrophy and treated with metreleptin, an improvement in steatosis and ballooning injury, as well as a reduction in liver volume and fat content, was observed [143]. However, further research in hepatic metabolism and metreleptin in FPLD is needed. A metabolomic analysis was also carried out, revealing that metreleptin may lead to changes in pathways involving branched-chain amino acid metabolism, fatty acid oxidation, protein degradation, urea cycle, tryptophan metabolism, nucleotide catabolism, vitamin E and steroid metabolism in 19 subjects with lipodystrophy (eight with FPLD). Moreover, the use of 3D stereophotogrammetric imaging in eight patients with lipodystrophy treated with metreleptin (four with FPLD2 and two with FPLD3) revealed the loss of facial soft tissue volume, although this was not visible to the naked eye in most cases [144]. 

A reliable cut-off point to select lipodystrophy responders to metreleptin has not been identified [145]. In fact, when comparing patients with FPLD2 with severe hypoleptinaemia (serum leptin <7th percentile of normal) vs. those with moderate hypoleptinaemia (serum leptin in 7–20th percentiles) leptin replacement therapy was found to be equally effective in reducing serum and hepatic triglyceride levels, with no improvement in hyperglycaemia [140]. Leptin treatment also resulted in similar metabolic improvements when comparing patients with FPLD2 and FPLD3 [146]. Furthermore, a recent study has shown evidence suggesting that patients with generalised or partial lipodystrophy treated with metreleptin (20 of 103 had FPLD2) can potentially reduce the risk of mortality (estimated 65% decrease in mortality risk) despite greater disease severity in treated patients in comparison with metreleptin-naïve patients [131].

### 3.7. Future Perspectives

In recent years we have been aware of certain advances regarding the pathogenetic mechanisms, clinical diagnosis and treatment of FPLD2. Regarding the aetiopathogenesis of the disease, although several theories have been proposed over the years (structural theory, gene expression theory, the theory of the alteration in cell proliferation/differentiation, the theories of prelamin A toxicity and failure of adipose tissue browning), new advances are being made in the field of epigenomics. Thus, one of the hypotheses being raised is that the previously-mentioned variability in clinical expressivity of this disease could be conditioned by certain epigenetic marks. In fact, there are studies that suggest that the mechanisms responsible for the loss of adipose tissue may be closely related to certain miRNAs and certain changes in histones. Specifically, the role of dynamic chromatin remodelling on adipogenesis has recently been described, and more specifically, how variants in *LMNA* related to FPLD2 can modulate certain epigenetic marks such as miR-335, altering the processes of adipocyte differentiation [147,148]. Other pathogenetic mechanisms that may be relevant are the dysregulation of autophagy and its relationship with adipocyte transdifferentiation processes [19]. The differential expression in subcutaneous adipose tissue of certain lncRNAs, such as the *HOTAIR* gene, and their fundamental role in adipocyte differentiation processes [149] undoubtedly raises new questions when it comes to an understanding of how variants in different genes lead to similar adipose phenotypes.

Nevertheless, the initial diagnosis of the disease remains merely clinical. For this reason, applications (such as LipoDDx^®^) are being developed that enable the identification of different lipodystrophy subtypes, such as FPLD2, and assist in their diagnosis with a remarkable degree of efficiency [150]. Even so, the algorithms of these applications are based on the physician’s ability to recognise certain signs and symptoms, which is not always the case due to a lack of knowledge and/or experience taking into account the rarity of these disorders. For this reason, efforts are also being made to develop software based on Artificial Intelligence, focused on the identification of lipodystrophy syndromes [151] using deep-learning techniques, with the ultimate goal of helping any physician to accurately diagnose these syndromes. In addition, also taking into account the rarity of FPLD2, only large and prolonged registries will make it possible to present a more objective picture of the disorder at all levels, which is why international cooperation in the research area is mandatory. For this purpose, in 2017, a European registry in the area of lipodystrophy (ECLip Registry) was created (ClinicalTrials.gov Identifier: NCT03553420) [152] from which upcoming results from ongoing prospective studies on lipodystrophy syndromes will be obtained.

As far as treatment is concerned, as previously indicated, the use of metreleptin in FPLD2 may be beneficial in selected cases for the control of metabolic and hepatic complications, although it is necessary to wait for the results of ongoing clinical trials. The same can be said for other molecules still in the experimental phase, such as leptin receptor agonist antibodies (ClinicalTrials.gov Identifier: NCT05088460).

## 4. Conclusions

FPLD2 is a rare and underdiagnosed disorder, especially in men. However, beyond the undeniable social stigma due to changes in the phenotype, it entails serious complications that worsen the quality of life and may reduce life expectancy. Its diagnosis, which is merely clinical, is not difficult in women among trained physicians. The marked loss of fat in the limbs and buttocks from puberty, associated with an accumulation of adipose tissue on the face and neck, together with muscle hypertrophy and stigmata of insulin resistance, should alert the clinician to this entity and, therefore, seek its molecular confirmation. This will also serve to identify potential carriers of these variants in *LMNA*, with special emphasis on men. In addition, Dunnigan disease can become a serious disorder with potentially lethal associated complications, requiring strict monitoring by multidisciplinary teams, including endocrinologists, cardiologists, hepatologists, neurologists, obstetricians, and psychologists. This type of approach can rarely be achieved without specialised reference centres or units. New advances in aetiopathogenesis, clinical diagnosis and therapeutic approaches have also been developed in recent years, although future research in this field through large registries is still needed.

## Figures and Tables

**Figure 1 cells-12-00725-f001:**
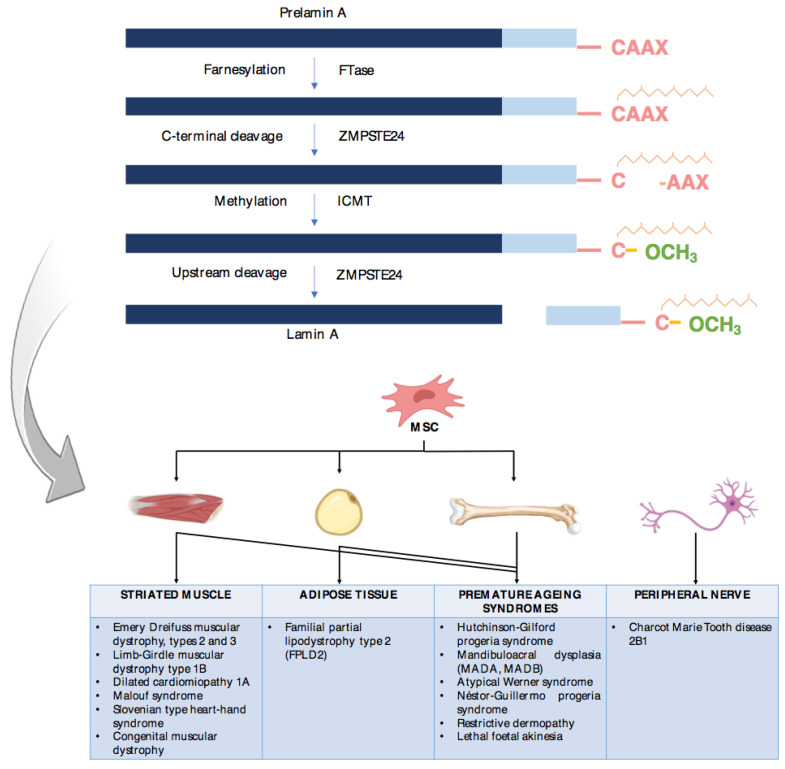
Lamin A processing pathway and laminopathies affecting mesenchymal tissues. Prelamin A is farnesylated, methylated and processed by ZMPSTE24, giving rise to the mature lamin A. Alterations in this lamin A processing pathway are responsible for several disorders known as laminopathies, which mainly affect mesenchymal tissues. In this figure, laminopathies are distributed according to the most characteristically affected mesenchymal tissue (adipose tissue in the case of type 2 familial partial lipodystrophy). MSC: mesenchymal stem cell.

**Figure 2 cells-12-00725-f002:**
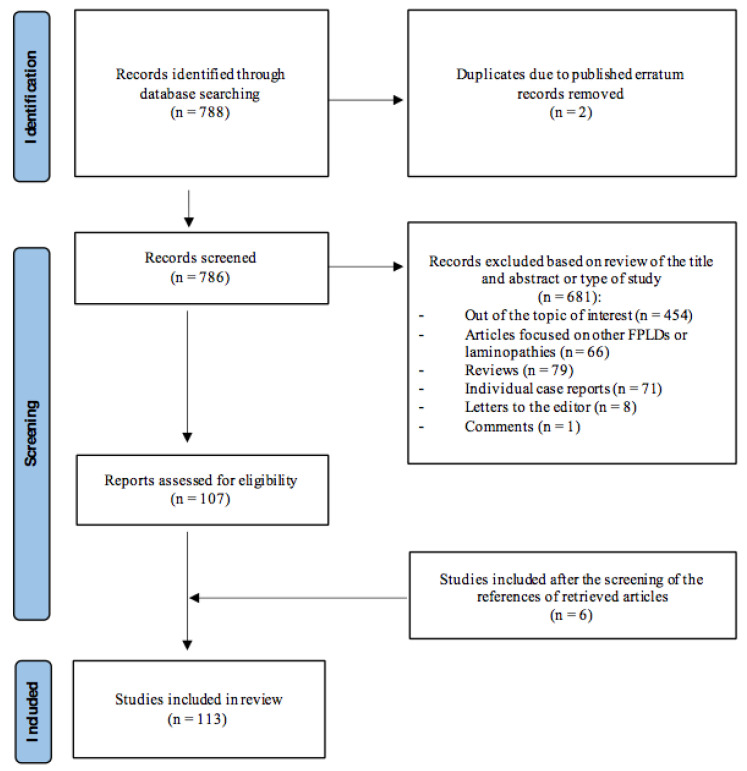
Search strategy and flow diagram of articles.

**Figure 3 cells-12-00725-f003:**
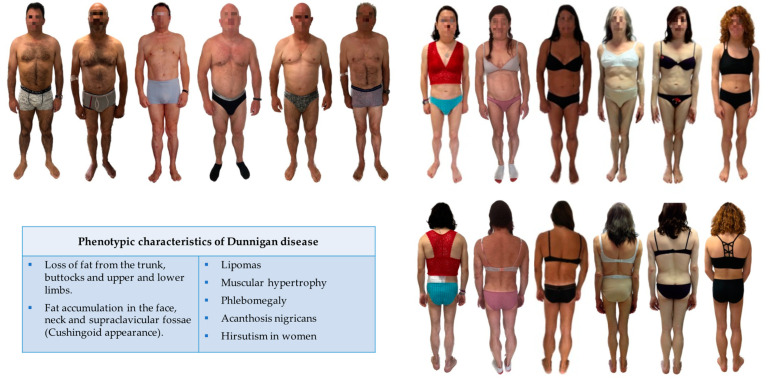
Clinical features of Dunnigan disease. While there is a loss of adipose tissue from the trunk, buttocks and limbs, fat excess in the face and neck and muscle hypertrophy is evident in women with FPLD2; in men, this characteristic phenotype is not so apparent.

**Figure 4 cells-12-00725-f004:**
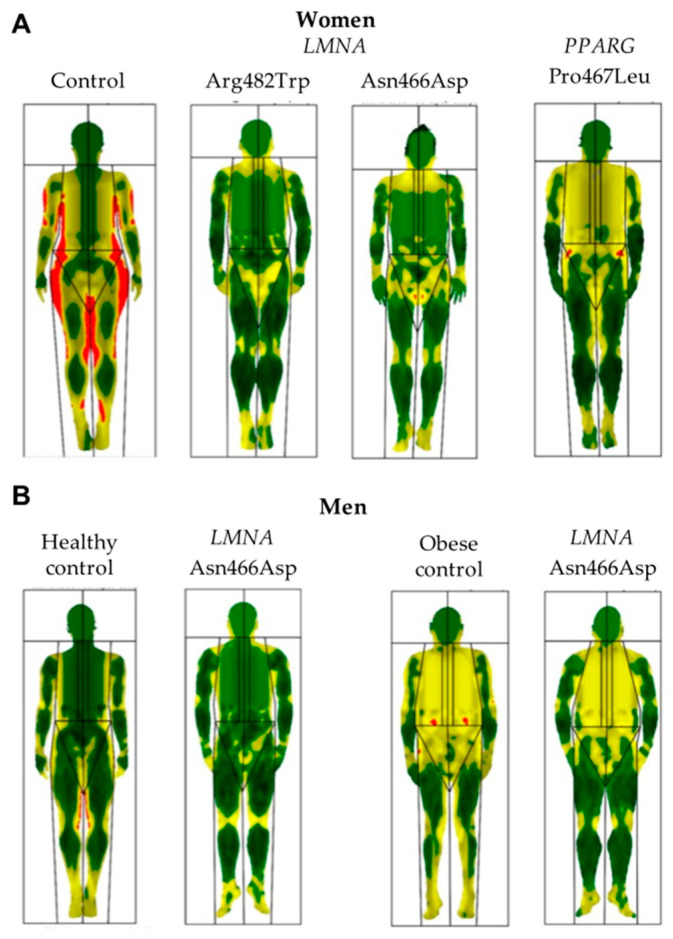
Comparative body composition determined by Dual-Energy X-ray Absorptiometry. Comparative, colour-mapped, total body composition scans via whole-body Dual-Energy X-ray Absorptiometry of patients with FPLD2, FPLD3 and non-lipodystrophic subjects. Green represents an area of low-level % fat (0–25%), yellow an area of medium-level % fat (25–60%), and red an area of high-level % fat (60–100%); (**A**) Less-pronounced fat loss can be observed in the woman with *PPARG*-associated FPLD in comparison with the women with FPLD2. Differences can be seen even when comparing different variants within exon 8 of the *LMNA* gene; (**B**) Differences in the distribution of adipose tissue are less evident in men with FPLD2 when compared with age and BMI-matched healthy or obese controls.

**Table 1 cells-12-00725-t001:** Clinical characteristics of FPLD2 and its differential diagnosis with other familial partial lipodystrophy syndromes.

FPLD2 Main Characteristics [4,5,6,13,14,15,16,17,18,19,20,21,22,23,24,25,26,27,28,29,30,31,32,33,34,35,36,37,38,39,40,41,42,43,44,45,46,47,48,49,50,51,52,53,54,55,56,57,58,59,60,61,62,63,64,65,66,67,68,69,70,71,72,73,74,75,76,77,78,79,80,81,82,83,84,85,86,87,88,89,90,91,92,93,94,95,96,97,98,99,100,101,102,103,104,105,106,107,108,109,110,111,112,113,114,115,116,117,118,119,120,121,122,123,124,125,126,127,128]		Differential Characteristics of other FPLD Syndromes [63,64,65,66,67,68,69,70,71,72,73,74,75,76,77,78,79,80,81,82,83,84,85,86,87]
Gene	*LMNA*	Unknown (FPLD1), *PPARG* (FPLD3), *PLIN1* (FPLD4), *CIDEC* (FPLD5), *LIPE* (FPLD6), *CAV1* (FPLD7), *AKT2*, *PCYT1A*, *ADRA2A*, *MFN2*.
Inheritance	AD (homozygous cases have also been described).	AR in FPLD5, FPLD6, *PCYT1A*- and *MFN2*-related FPLD
Onset of phenotype	Puberty in women, later in men.	- Birth (FPLD7) - Childhood-adulthood (other FPLD).
Abnormal fat distribution	- Loss of fat in the limbs, trunk and gluteal region. - Accumulation of fat in the face, neck, chin, axillae, interscapular area, labia majora and abdominal viscera. - Less apparent in men. - Atypical FPLD2 (non-codon 482 FPLD2): milder phenotype.	- FPLD1: accumulation of abdominal fat. - FPLD3: less severe loss of fat. - FPLD7: loss of fat in the face and upper body.
Clinical features	- Subcutaneous lipomas. - Muscular hypertrophy, mialgias. - Phlebomegaly. - Signs of insulin resistance (acanthosis nigricans, acrochordons). - Hyperandrogenism. - Hypertrophy of mons pubis, “Dunnigan sign”.	- FPLD1: KöB index > 3.477. - FPLD3: less prominent musculature, no phlebomegaly. - FPLD6: multiple lipomatosis. - FPLD7: congenital cataracts, progeroid features. - *PCYT1A*-related FPLD: short stature, muscular atrophy. - *MFN2*-related FPLD: lipomatous masses.
Analytical parameters	- Lower plasma leptin and adiponectin levels than controls, with no standardised cut-offs. - Generally increased TNF-α, IL-1β, IL-6 and IL-10 levels.	- *MFN2*-related FPLD: very low leptin levels. - FPLD6: elevated creatine kinase levels.
Main comorbidities	- Insulin resistance. - Type 2 diabetes mellitus. - Dyslipidaemia (high triglyceride levels, low HDL cholesterol). - Acute pancreatitis. - Hypertension. - Cardiovascular disease (cardiac hypertrophy, atrioventricular conduction defects, heart failure, early atherosclerosis, arrhythmias). - Liver disease (NAFLD, NASH). - Fertility problems. - PCOS. - Renal disease (proteinuria and progression to renal failure). - Anticipation phenomenon (diabetes, hypetriglyceridaemia).	- FPLD3: earlier and more severe metabolic complications. Early hypertension. - FPLD6: auto-fluorescent drusen-like retinal deposits. Muscular dystrophy in some patients. - *MFN2*-related FPLD: peripheral axonal neuropathy.

In order to demonstrate a comprehensive comparison between FPLD2 and other FPLD subtypes, references [63,64,65,66,67,68,69,70,71,72,73,74,75,76,77,78,79,80,81,82,83,84,85,86,87], related to FPLD subtypes other than Dunnigan disease, have been added in addition to the retrieved articles from the database search and reference screening. AD: autosomal dominant; AR: autosomal recessive; NAFLD: non-alcoholic fatty liver disease; NASH: non-alcoholic steatohepatitis; PCOS: polycystic ovary syndrome.

## Data Availability

No new data were created or analysed in this study. Data sharing is not applicable to this article.

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
