# Peer review of "Clinical Spectrum of LMNA-Associated Type 2 Familial Partial Lipodystrophy: A Systematic Review"

_cells, 2023, doi:10.3390/cells12050725_

Round 1

Reviewer 1 Report

This is an interesting review article focused on the clinical spectrum of LMNA-associated Type 2 Familial Partial 2. The review presents many valuable data. However, there are some important concerns to be addressed.

 1.      Title of the review is not in line with rationale (last paragraph in Introductioin section) and some part of text (3.7. Other laminopathies, Tables, etc.). It is not clear whether this review is about LMNA-associated FPLD2 or all FPLD syndromes? Thus, the topic should be clarify, and accordingly create a table with main characteristics of FPLD in population (from 113 studies included in this systematic review).

2.      Table 1 is inserted in subsection 3.4. Comorbidities and organ abnormalities. However, it is not clear what is the subject of this review!? LMNA-associated FPLD2 or all FPLD syndromes?

3.      3.4.3. Lipid metabolism and pancreatitis. This subsection should be improved. There are only two sentences regarding pancreatitis, inserted in the middle of the text regarding lipid metabolism.

4.      3.4.9. Other organ abnormalities. The First sentence from this subsection should be moved to subsection 3.4.4. Cardiovascular disease.

5.      3.7. Other laminopathies. This subsection, including Table 2., seem completely redundant and out of theme of this review. In addition, the position of this subsection is inadequate.

6.      Subsection regarding gender differences in FPLD2 should be added.

7.      Future perspective should be added.

8.      The Conclusion should be substantially improved, currently read more like summary of article.

Author Response

This is an interesting review article focused on the clinical spectrum of LMNA-associated Type 2 Familial Partial 2. The review presents many valuable data. However, there are some important concerns to be addressed.

  1. Title of the review is not in line with rationale (last paragraph in Introductioin section) and some part of text (3.7. Other laminopathies, Tables, etc.). It is not clear whether this review is about LMNA-associated FPLD2 or all FPLD syndromes? Thus, the topic should be clarify, and accordingly create a table with main characteristics of FPLD in population (from 113 studies included in this systematic review).

R: We appreciate the constructive comments of the reviewer. The purpose of this manuscript is to carry out a systematic review regarding the clinical characteristics, associated comorbidities and therapeutic approach to FPLD2. We believe that the differential diagnosis, which is eminently clinical, with other subtypes of familial partial lipodystrophy can be helpful to clinicians, although the definitive diagnosis will always be molecular. We have modified Table 1 to highlight the main characteristics of FPLD2 from the studies included in the review as suggested. In addition, we understand that it is important to highlight some unique clinical characteristics of other FPLD subtypes (which are phenotypically very similar to Dunnigan disease), and for this reason we have included in the new table only what is essential to differentiate them, in order to help the clinician to approach the differential diagnosis. Regarding the "Other laminopathies" section, since this manuscript has been submitted for publication in the Topical Collection "Lamins and Laminopathies", we had thought that a reference to other lipodystrophy syndromes associated with variants in the LMNA gene could be useful to underscore the phenotypic heterogeneity of laminopathies. However, we understand the reviewer's objections and we have removed this section from the new version of the manuscript. Consequently, the last paragraph of the Introduction section and the end of the Abstract have also been modified.

  1. Table 1 is inserted in subsection 3.4. Comorbidities and organ abnormalities. However, it is not clear what is the subject of this review!? LMNA-associated FPLD2 or all FPLD syndromes?

R: Table 1 has been modified, examining the clinical characteristics and associated comorbidities of FPLD2 in more depth. In this new table, only specific differential characteristics of other types of FPLD have been included to facilitate the differential diagnosis of FPLD2 given the great clinical similarity of these familial partial lipodystrophies.

  1. 4.3. Lipid metabolism and pancreatitis. This subsection should be improved. There are only two sentences regarding pancreatitis, inserted in the middle of the text regarding lipid metabolism.

R: This subsection has been improved. Please see the text marked with change tracking, lines 360-373.

  1. 4.9. Other organ abnormalities. The First sentence from this subsection should be moved to subsection 3.4.4. Cardiovascular disease.

R: Thank you for this suggestion. This sentence has been moved accordingly to subsection “3.4.4. Cardiovascular disease”in the new version of the manuscript (see text marked with change tracking, lines 408-411)

  1. 7. Other laminopathies. This subsection, including Table 2., seem completely redundant and out of theme of this review. In addition, the position of this subsection is inadequate.

R: This subsection has been removed in the new version of the manuscript.

  1. Subsection regarding gender differences in FPLD2 should be added.

R: Although there are hardly any studies specifically focused on identifying the differences between genders in FPLD2, we agree with the reviewer that it is especially relevant to emphasise these distinctions. For this reason, new comments in this sense have been added throughout the text, in addition to those previously written, in each corresponding subsection, maintaining the established structure of “prevalence – clinical characteristics – body composition – comorbidities and organ abnormalities – mortality – treatment”. Please, see lines 160-162 (onset of phenotype), 167-172 (lipodystrophy phenotype and clinical diagnosis), 344-346 (diabetes), 365-347 (lipid metabolism and pancreatitis), 391 and 408-411 (atherosclerosis and hypertension).

  1. Future perspective should be added.

R: We fully agree that a new section related to future perspectives will help to put the finishing touch to the article. Thus, a new section in this regard has been included in the new version of the manuscript. In this section we highlight the importance of new pathogenetic mechanisms that have been discovered in recent years, in addition to certain innovations for clinical diagnosis and a brief treatment-oriented paragraph.

  1. The Conclusion should be substantially improved, currently read more like summary of article.

R: We agree with the reviewer’s comment. A new conclusion has been included in the new version of the manuscript.

Reviewer 2 Report

This is another review of the fascinating conditions coming under the heading of lipdystrophy  by one of the two premier groups in the field.   Although the topic has been many times reviewed in the literature, it is always valuable to have continued documentation. What would be useful is to add something new and different from what is already written.   Clearly, a discussion of gene associated biochemical pathways would be gratifying, but it is not clear whether any progress has been made on this research.

In the alternative, perhaps the authors could discuss the history of discovery of the fascinating genetic research on the lipodystrophies.

Author Response

This is another review of the fascinating conditions coming under the heading of lipdystrophy  by one of the two premier groups in the field.   Although the topic has been many times reviewed in the literature, it is always valuable to have continued documentation. What would be useful is to add something new and different from what is already written.  Clearly, a discussion of gene associated biochemical pathways would be gratifying, but it is not clear whether any progress has been made on this research.

In the alternative, perhaps the authors could discuss the history of discovery of the fascinating genetic research on the lipodystrophies.

R: The kind comments of the reviewer are much appreciated. Some advances regarding the pathogenetic mechanisms of this disorder have been made in recent years and, although it is outside the scope of this review, we have included them in the new section "Future perspectives".

Reviewer 3 Report

The manuscript by Fernandez-Pombo et al., entitled ‘Clinical spectrum of LMNA-associated type 2 familial partial lipodystrophy. A systematic review’ is an updated systematic review (based on carefully selected 113 articles) that summarizes the very complex clinical manifestations of FPLD2. Their findings identified the clinical features of FPLD2, comorbidities, and current treatments. Very interestingly, the authors also presented a comprehensive comparison between FPLD2 and other lipodystrophy-associated laminopathies.

In general, the manuscript is amenable to read and well-structured. I have only the following minor comments:

- In table 2, given that the information gathered is related to lipodystrophy-associated laminopathies, the column ‘Dysfunction’ could be removed since the information is the same in all syndromes.

- Table 1 and 2 are very useful and gathered useful information to compare FPLD2 with other lipodystrophy syndromes. However, given that the aim of the manuscript is to summarizes the very complex clinical manifestations of FPLD2, a table with a nice summary of the FPLD2 phenotypic characteristics and/or comorbidities would make the review more impactful.

 - In the figure 1 legend, the term Dunnigan disease appear for the first time in the manuscript. The term should be presented in the main text before.

Author Response

The manuscript by Fernandez-Pombo et al., entitled ‘Clinical spectrum of LMNA-associated type 2 familial partial lipodystrophy. A systematic review’ is an updated systematic review (based on carefully selected 113 articles) that summarizes the very complex clinical manifestations of FPLD2. Their findings identified the clinical features of FPLD2, comorbidities, and current treatments. Very interestingly, the authors also presented a comprehensive comparison between FPLD2 and other lipodystrophy-associated laminopathies.

In general, the manuscript is amenable to read and well-structured. I have only the following minor comments:

- In table 2, given that the information gathered is related to lipodystrophy-associated laminopathies, the column ‘Dysfunction’ could be removed since the information is the same in all syndromes.

R: This table has been removed following the advice of reviewer 1.

- Table 1 and 2 are very useful and gathered useful information to compare FPLD2 with other lipodystrophy syndromes. However, given that the aim of the manuscript is to summarizes the very complex clinical manifestations of FPLD2, a table with a nice summary of the FPLD2 phenotypic characteristics and/or comorbidities would make the review more impactful.

R: We agree with the reviewer in placing more emphasis on the phenotypic characteristics and comorbidities of FPLD2 in Table 1 given the aim of the article. Therefore, this table has been modified.

 - In the figure 1 legend, the term Dunnigan disease appear for the first time in the manuscript. The term should be presented in the main text before.

R: The term Dunnigan disease has now been presented in the main text at the first moment in which FPLD2 is named (see line 40), instead of in line 58. The legend of figure 1 has also been changed. We appreciate the suggestion.

Round 2

Reviewer 1 Report

Authors have adequately addressed all concerns raised by this reviewer. I have no further comments.